# Evaluation of Shiitake Mushroom (*Lentinula edodes*) Supplementation on the Blood Parameters of Young Thoroughbred Racehorses

**DOI:** 10.3390/ani12223212

**Published:** 2022-11-19

**Authors:** Maria Soroko-Dubrovina, Wanda Górniak, Paulina Zielińska, Aleksander Górniak, Nina Čebulj-Kadunc, Mariusz Korczyński

**Affiliations:** 1Institute of Animal Breeding, Wroclaw University of Environmental and Life Sciences, Chelmonskiego 38C, 51-630 Wroclaw, Poland; 2Department of Automotive Engineering, Mechanical Faculty, Wroclaw University of Science and Technology, Na Grobli 13, 50-421 Wroclaw, Poland; 3Department of Surgery, Faculty of Veterinary Medicine, Wroclaw University of Environmental and Life Sciences, Plac Grunwaldzki 51, 50-366 Wroclaw, Poland; 4Institute of Preclinical Sciences, Veterinary Faculty, University of Ljubljana, Gerbičeva 60, 1000 Ljubljana, Slovenia; 5Department of Animal Nutrition and Feed Management, Wroclaw University of Environmental and Life Sciences, Chelmonskiego 38C, 51-630 Wroclaw, Poland

**Keywords:** horses, shiitake mushroom, Thoroughbred, health, nutrition, supplement

## Abstract

**Simple Summary:**

Shiitake mushrooms (*Lentinula edodes*) are a valuable source of many important nutrients, vitamins and minerals. However, there are currently no standard protocols for testing the efficacy of shiitake mushrooms in equine nutritional supplementation. In total, 20 Thoroughbreds were divided into either the supplemented group or the control group. The supplemented group was given 30 g of *L. edodes* daily for four months. One blood sample was collected four times from each horse at four-week intervals. Our study showed differences between individual days of blood sampling in the supplement and control groups. Blood hematology showed that supplementation had an effect on monocytes. In a biochemical analysis, the shiitake mushroom affected the level of alkaline phosphatase, glucose, aspartate aminotransferase, lactic acid and cholesterol levels measured in the individual days of blood sampling. Our study is the first to demonstrate the influence of shiitake mushroom supplementation on blood parameters in young racehorses undergoing regular training. The greatest effects of supplementation were found for monocytes and alkaline phosphatase. Further studies are needed to show the influence of supplementation with shiitake mushrooms in larger groups of horses over a longer period.

**Abstract:**

The aim of this study was to evaluate the effect of shiitake mushroom (*Lentinula edodes*) supplementation on the hematology and biochemical blood parameters of young Thoroughbred racehorses. The study was conducted with 20 horses divided into two groups: the supplemented and the control group. The supplemented group was given 30 g of *L. edodes* daily for four months. One blood sample was collected four times from each horse at four-week intervals. The hematology analysis in the supplemented group showed a higher level of monocytes at day 56 when compared to the control group (*p* = 0.000986). Biochemical analysis showed that alkaline phosphatase is most sensitive to shiitake mushroom supplementation, with statistically significant lower levels in supplemented group compared to the control group on all individual days of blood sampling. It was also found that supplementation had an effect on the decrease of glucose levels on days 28 (*p* = 0.009109) and 56 y (*p* = 0.025749), on reduction aspartate aminotransferase level on day 56 (*p* = 0.017258) and a decrease of lactic acid on day 28 of sampling (*p* = 0.037636). Cholesterol levels decreased consistently in all individual days of blood sampling. Further studies are needed to show the influence of supplementation with shiitake mushroom in larger groups of horses over a longer period.

## 1. Introduction

Shiitake mushrooms (*Lentinula edodes*) have remarkable health properties that have been used in Far Eastern medicine for centuries. In Asia, shiitake mushrooms have been cultivated for more than 1000 years and are now the third most cultivated mushroom variety in the world. Due to their health-promoting properties and easily digestible ingredients, they are referred to as the “elixir of life” in Asian countries [1,2]. These mushrooms, used as “functional foods”, have a long history in oriental folklore in the treatment of cancer, viral diseases, cardiovascular diseases, obesity, diabetes, respiratory diseases, liver disorders, fatigue and weakness [3]. Shiitake mushrooms are also called “medicinal mushrooms” because of their properties and their use as a dietary supplement to support patients during oncological therapy and radiotherapy [1,4,5].

Shiitake mushrooms are a source of two types of polysaccharide medicine: β-D-glucans (extracted from the mycelium) and fungal bodies [6]. Both components activate the immune system [7,8], have antioxidant properties [9] and significantly influence the regeneration of the body in studies based on humans and fish [10,11]. They also lower cholesterol levels, a process brought about by eritadenine, which can reduce low-density lipoprotein cholesterol by increasing the excretion of cholesterol through the faeces in humans [1]. Eritadenine is also a source of fatty acids with a high content of amino acids [12]. Shiitake mushrooms also contain significant amounts of vitamins, especially C, B1, B2 and B12 (niacin), which reduce stress responses and have calming effects [13]. They contain the highest amount of vitamin D among plant foods, with a presence of calcium [14,15].

In contrast to humans, the effects of shiitake mushrooms on animal health are scarce and mostly limited to laboratory animals such as rats [16] and mice [17] or poultry [18,19,20,21,22]. In rats, shiitake supplementation affected the growth of specific gut microbes, e.g., *Clostridium* and Bacteroides spp., compared to the control groups, and manipulation of the gut microbiota through the administration of *L. edodes* managed dyslipidemia [16]. Research on mice indicated significantly lowered serum total cholesterol, LDL cholesterol and triglycerides after mushroom supplementation, which could help in the regulation of lipid metabolism [17], and which was also suggested by the authors of a recent study of shiitake effects on dogs [23]. Several studies have confirmed the health-promoting effects of supplementation with shiitake mushroom on immune response modulation and stress reduction in poultry [18] and antibacterial, antiviral and antiseptic activity in chickens [19,20,21], as well as growth performance in broilers [22].

In our previous study on horses, supplementation with shiitake mushroom significantly influenced several hematology and blood biochemistry parameters and confirmed the beneficial effects of mushroom supplementation on horses [24]. However, there currently are no standard protocols to verify the efficacy of shiitake mushroom in equine nutritional supplementation. The lack of studies on the recommended daily dose of shiitake mushrooms in relation to blood parameters has limited research in this field. This requires more focused research on different daily doses and a standardized study group of horses to limit the physiological effects on blood parameters. The aim of the present study was therefore to investigate the effects of dietary supplementation with shiitake mushrooms (*L. edodes*; 30 g per day over a period of four months) on the hematological and biochemical blood parameters of young Thoroughbred racehorses in regular training.

## 2. Materials and Methods

Ethical approval was obtained from the Local Ethical Committee for Animal Experiments in Wroclaw, prior to data collection in June 2019 (resolution no. 036/2019/P1).

### 2.1. Animals

The study was conducted on a group of 20 three- to four-year-old Thoroughbred racehorses, with an average body weight of 454 ± 22 kg. The horses were clinically healthy and in good condition. They all had a similar fitness level and were trained daily for flat races at Partynice Racecourse (Poland). The daily training duration was the same for all horses. The animals were housed in individual boxes on straw, without access to pastures. All the boxes measured about 12.25 m^2^ and were cleaned six times a week. The animals were dewormed and vaccinated one month before the experiment according to standard veterinary practice. They were all fed a basal diet of oats and concentrates (muesli) (Oat Balancer Mix, Baileys Horse Feed, Braintree, Essex, UK—ingredient composition: micronized barley, soybean flakes, maize and soya, molasses, micronized wheat, distillers’ rains, soya oil, hulls of soybeans, cooked linseed, ScFOS (Digest Plus prebiotic) vitamins and minerals and roughage (meadow hay), in accordance with the requirements for racehorses, based on the Nutrient Requirements of Horses [25]. Water and mineral salt (Lisal, Klodawa, Poland) were provided as needed. Horses were fed three times a day, at 5.30 a.m., 12.00 p.m. and 5.00 p.m. The diet was followed for 6 months before the first samples were taken and continued throughout the study.

### 2.2. Experimental Procedure

The lyophilizate of shiitake mushrooms used in the study was produced on the Ecological Farm for the Cultivation of Chinese Mushrooms in Kalisz, Poland. The shiitake mushroom powder was analyzed and contained the following: dry weight—90.77%, ash—6.71 g/kg; protein—25.71 g/kg; fat—1.04 g/kg; fiber—9.05 g/kg; Mg—1.11 g/kg; Ca—0.28 g/kg; F—1.71 g/kg; N—19.95 g/kg; K—19.95 g/kg; Cu—6.98 mg/kg; Mn—17.41 mg/kg; Fe—82.75 mg/kg; Zn—66.64 mg/kg; and Se—871.85 ug/kg. The powder was granulated with additives so that 100 g of granulate contained 30 g of dried shiitake mushroom, 50 g of barley, 10 g of wheat bran, 4 g of corn, 4 g of lucerne and 2 g of molasses.

The horses participating in the study were randomly divided into two groups: the first, the supplemented group (G0) consisting of ten horses (five aged three years, five aged four years), and the second group was the control group (G1), which also included ten horses (five aged three years, five aged four years). The supplemented group received an addition of 100 g granulate containing 30 g of shiitake mushrooms once daily (during 12:00 feeding) throughout the entire study period (from March until June). The control group was given granules without the addition of *L. edodes*.

### 2.3. Blood Collection and Analysis

The protocol for blood sampling was the same as in our previous study [24]. Blood samples were collected four times, four weeks apart, starting on day 1 of treatment, followed by day 28, day 56 and day 84.

Blood was collected at rest, before morning feeding, by puncturing of the external jugular vein (*vena jugularis externa*) with a sterile BD Vacutainer^®^ system including 20 G 1/2” needles. K2-EDTA tubes were used for hematological analysis, and plain tubes for serum biochemistry and fluoride tubes (sodium fluoride 15 mg/mL/EDTA 3.0 mg/mL) for determination of to assess lactic acid concentration (lactic acid, mmol/L) (BD Vacutainer Systems, Plymounth, UK). All blood samples were examined within a maximum period of 1.5 h after collection.

Blood hematological analyses were performed using a Sysmex XN-1000 hematology analyzer (Sysmex America, Inc., Lincolnshire, Illinois, IL, USA). The following parameters were assessed: white blood cell count (WBC, 10^9^/L), absolute and differential distribution of neutrophils (NEU, 10^9^/L, %), lymphocytes (LYM, 10^9^/L, %), monocytes (MONO, 10^9^/L, %), eosinophils (EOS, 10^9^/L, %), and basophils (BASO, 10^9^/L, %), red blood cell count (RBC, 10^12^/L), hemoglobin concentration (HGB, mmol/l), hematocrit (HCT, L/L,%), mean corpuscular volume (MCV, fL), mean corpuscular hemoglobin (MCH, fmol), mean corpuscular hemoglobin concentration, (MCHC, mmol/L) and platelet count (PLT, 10^12^/L).

The plain tubes were centrifuged (2800 rpm for 5 min) using a benchtop Rotanta 460 centrifuge (Andreas Hettich GmbH & Co. KG, Tuttlingen, Germany), with the serum aspirated. The levels of the following biochemical parameters were estimated: albumin (g/L), alkaline phosphatase (AP, U/L), aspartate aminotransferase (AST, U/L), total protein (g/L), total bilirubin (μmol/L), chlorides (mmol/L), cholesterol (mmol/L), creatine kinase (CK, U/L), phosphorus (P, mmol/L), glutamate dehydrogenase (GLDH, U/L), glucose (GLUC, mmol/L), gamma-glutamyl transferase (GGTP, U/L), creatinine (μmol/L), lactate dehydrogenase (LDH, U/L), magnesium (Mg, mmol/L), urea (mmol/L), potassium (K, mmol/L), sodium (Na, mmol/L), triglycerides (TGL, mmol/L), calcium (Ca, mmol/L), globulins (g/L) and albumin/globulin ratio (mmol/L). All the parameters were determined using a AU680 clinical chemistry analyzer (Backman Coulter Inc., Brea, CA, USA). Sysmex (Sysmex America, Inc., Lincolnshire, IL, USA) and Backman (Backman Coulter Inc., Brea, CA, USA) reagents, calibrators and standards were used for all the measurements. Only GLDH concentrations were determined using Randox diagnostic veterinary reagents (Randox, Crumlin, UK).

### 2.4. Data Analysis

All tests were performed with a significance level of α = 0.05. The Shapiro–Wilk and Kolmogorov–Smirnov tests of normality proved that the measurement did not originate from a Gaussian distribution, so non-parametric tests were used. The differences between the supplemented group (G0) and the control group (G1) were tested using the Mann–Whitney U significance test. In the following, each blood sampling was considered separately. Only the differences between the two groups were compared and the groups as such were considered independent. A Friedman ANOVA test was used to test the significance of differences in blood parameters measurements, between the G0 and G1 groups. For significant parameters, a post-hoc (Conover–Iman) test was used to test for the differences between individual days of sampling. Statistica software (v. 13.3, StatSoft Inc., Tulsa, OK, USA) was used for all calculations.

## 3. Results

### 3.1. Blood Hematological Analysis

The results of the comparisons of the hematological analysis between the supplemented and control groups are presented in Table 1. Statistically significant differences were found for MONO and MONO% at day 56. Considering the *p*-value of MONO% in this session, there was a significant difference between the supplemented group and the control group.

Friedman’s ANOVA was used to test the influence of shiitake mushroom supplementation throughout the test period. The same horses were compared across different days of blood sampling and, as such, the variables were considered as dependent. Significant differences were found in ten parameters between the days of blood sampling (Table 2). The largest difference was found for MCV (*p* = 0.000099).

Friedman’s ANOVA proved that parameters differed between sampling days but there was no indication between which days. To determine the differences between the two groups, the median parameters of the hematological blood analysis for the supplemented and control groups are shown in Figure 1. No clear patterns were observed for the blood parameters presented, nor were there any significant differences between the groups studied.

### 3.2. Blood Biochemical Analysis

An analysis of the differences in biochemical factors is shown in Table 3. Statistically significant differences in AP were found between the two groups, on all days of sampling. The largest differences were found on days 28 and 56. For glucose, there were statistical differences on days 28 and 56. Considering the *p*-value of glucose on day 1, it seems to be close to statistical validity. In addition, AST and lactic acid only differed significantly during individual days of blood sampling.

Differences were found between biochemical values on each day of blood sampling for most parameters (Table 4). The greatest difference was found for LDH, which increased significantly on day 56 and decreased on day 84 in both groups. The opposite trend was noted for albumin, which decreased on day 56 and increased on day 84 in the supplemented group. In addition, cholesterol levels decreased throughout the experiment, whereas GGTP increased. Furthermore, differences were found between the days of blood sampling in both the research and control groups.

Considering the median parameters of the blood biochemical analysis for all days of blood sampling (Figure 2), a significant variability can be observed. Significant variations in lactic acid, CK and AST and albumin values were observed in the control group, compared to the supplemented group. For the horses in the supplemented group, the median value of the lactic acid was at a stable low level compared to the control group, without a large jump on day 84. The median cholesterol in G0 decreased steadily and remained constant in G1. The levels of bilirubin, CK and AST were lower in G0 than in G1. A closer comparison of the *p*-values of these groups suggests that the control group differs more between days of blood collection than the supplemented group.

## 4. Discussion

The present study was conducted to determine the effects of *L. edodes*, added to commercial feed, on the hematological and biochemical blood parameters of young Thoroughbred racehorses in regular training. The results indicated single statistical differences among training days between the supplemented and control groups.

Differences in hematological parameters between the two groups were only found for MONO at day 56, where the values for MONO were within normal ranges for both the experimental and control groups [26]. Monocytes are a critical component of the innate immune system, capable of rapid transfer to sites of inflammation, phagocytosis, pathogens or allergens [27]. The results of previous studies conducted on different species of fish and broilers suggest that diets supplemented with *L. edodes* mushrooms enhance immune response [28,29]. Interestingly, in our previous study based on riding horses supplemented with shiitake mushrooms, they had also manifested elevated MONO values [24].

Concerning blood biochemical parameters, the supplemented groups of horses showed a statistically significant reduction in AP values on all days of sampling compared to the control group, whereas values for AP were within normal ranges for two groups [30]. The greatest differences were found on days 28 (value of median of 289 U/L) and 56 (value of median of 204 U/L). Furthermore, the overall decrease of AP from day 28 to day 84 was approximately 41%. The AP medians of the control group did not exhibit this tendency. Interestingly, we found very similar results in our previous study, where horses supplemented with shiitake mushrooms also had lower levels of AP [24]. AP enzyme production is routinely used as a marker of liver function. It was found that different types of infectious and inflammatory diseases caused an increase in AP, which is probably a secondary phenomenon in different types of infectious and inflammatory liver diseases and is related to increased hepatocellular metabolism, most likely caused by a general stress on the organism under different disease conditions [31,32].

The results of the present study showed significant differences in glucose between the two groups on the two days of blood sampling. Similar results were presented in our previous study [24], where horses supplemented with *L. edodes* had lower glucose concentrations. In particular, the biologically active polysaccharides (β-glucans) contained in the mushrooms can restore the pancreatic tissue function, causing better utilization of insulin, thus preventing high blood glucose levels [33]. The present study also indicated a significant decrease in AST in the supplemented group, but only once. AST is a liver enzyme that is released into the bloodstream when liver dysfunction occurs. A study with a group of mice diagnosed with hepatitis showed a significant improvement in liver histology, indicated by a decrease in AST serum levels, after the mice were fed mushroom extracts [34]. In the present study, a significant difference was also found between the two groups in lactic acid levels. Blood lactate is a metabolic end-product occurring during glycolysis of carbohydrates under anaerobic conditions, serving as a key energy source during intense exercise [35]. Lactic acid that accumulates in the blood during exercise lowers muscle pH, which can lead to acidosis [36].

Our study showed a tendency to lower cholesterol levels in the supplemented group, which is consistent with studies in school riding horses [24]. Previous studies have reported that eritadenine, a hypocholesterolemic factor isolated from shiitake mushrooms, suppresses the biosynthesis of cholesterol in the liver and lowers plasma cholesterol concentrations in rodents [37,38]. Another study based on rats indicated that administration of polysaccharides from shiitake mushrooms significantly reduced serum totalcholesterol level in rats [39], whereas a study based on healthy dogs given shiitake mushroom powder for four weeks indicated a decrease in total cholesterol levels [23].

Current studies indicate significantly lower levels of bilirubin in the supplemented group, as compared to the control group. Bilirubin has a tendency to evaluate during exercises due to intervascular hemolysis during exercises [40], or in response to increased erythrocyte fragility, functional disorders of bile duct [41] and damage to hepatocytes or functional disorders of the liver [42]. In our previous studies, we also observed bilirubin levels to be lower in the supplemented group with shitake mushrooms [24].

Studies in horses indicate an increase in albumin levels after acute exercise as well as during prolonged training schedule, most likely due to profuse sweating during exercise, causing a shift of fluid from the intravascular space to the interstitial and intracellular space. This shift between the fluid compartments of the body leads to a decrease in blood volume and hemoconcentration [43,44,45]. On the other hand, proteomic studies in horses suggest a decrease in albumin fragments after competition, returning to baseline levels after 48 h, which may also be related to exercise-induced damage to liver cell function and/or increased nutrient requirements during intense exercise [46]. There was a tendency for albumin to decrease in most of the supplemented groups, although a study using rainbow trout found no differences in albumin between the control and shiitake mushroom supplemented groups [47]. Similar results were found in a study on rats, where dietary supplementation with shiitake mushrooms had no effect on albumin levels [48]. In another study, the albumin concentration in the serum of rats was reduced by 10% after a shiitake mushroom diet [49].

Summarizing the general results of the blood biochemical parameters, the supplemented group of horses was characterized by statistically significant but smaller differences in biochemical parameters throughout the entire study period.

It should be noted that there were some limitations to the present study. These include the short duration of the study, the small number of horses, the presence of different environmental variables and the lack of previous research on this topic. These are obstacles that should be considered in future research. However, the findings of the present study suggest that shiitake mushrooms have many health-promoting effects. Consequently, a larger study in horses with many more health-related investigations is warranted.

## 5. Conclusions

In conclusion, oral administration of *L. edodes* for four months at a daily dose of 30 g in young racehorses exercising regularly indicates an effect on their blood parameters in comparison to the untreated horses. The greatest effects of supplementation were found for monocytes and alkaline phosphatase. Further studies are needed to identify the causal effects of shiitake mushroom supplementation on various hematological parameters on larger groups of horses over a longer period of time.

## Figures and Tables

**Figure 1 animals-12-03212-f001:**
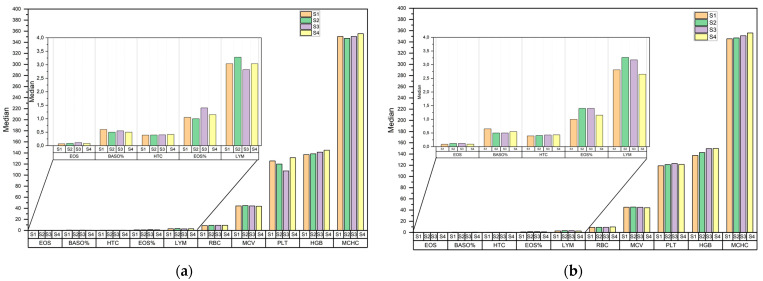
Hematological blood analysis median values for the: (**a**) supplemented group (G0); (**b**) control groups (G1); day 1 (S1), day28 (S2), day 56 (S3), day 84 (S4).

**Figure 2 animals-12-03212-f002:**
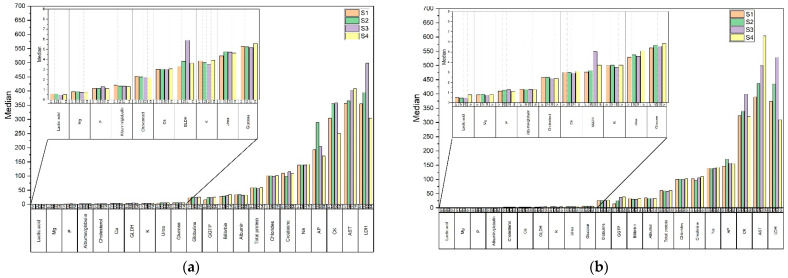
The median values of biochemical blood analysis for: (**a**) supplemented group (G0); (**b**) control group (G1); day 1 (S1), day28 (S2), day 56 (S3), day 84 (S4).

**Table 1 animals-12-03212-t001:** *p*-values representing differences between the supplemented (G0) and control group (G1) in a blood hematological analysis.

Parameter, Unit	Day 1 G0 vs. G1	Day 28 G0 vs. G1	Day 56 G0 vs. G1	Day 84 G0 vs. G1
WBC, 10^9^/L	0.344705	0.969839	0.570606	0.623177
NEU, 10^9^/L	0.495968	0.850107	0.909688	0.212295
NEU%	0.969850	0.969850	0.850107	0.140168
LYM, 10^9^/L	0.570751	0.909722	0.384674	0.384674
LYM%	0.596425	0.596564	0.733634	0.241145
MONO, 10^9^/L	0.255945	0.103590	0.037134	0.272496
MONO%	0.074563	0.103069	0.000986	0.211089
EOS, 10^9^/L	0.539499	0.288100	0.969679	0.381614
EOS%	0.878922	0.136421	1.000000	0.320046
BASO, 10^9^/L	0.907623	0.722023	1.000000	0.691548
BASO%	0.849094	0.969303	0.784078	0.412192
RBC, 10^12^/L	0.650025	0.622915	0.820397	0.705246
HGB, 10^12^/L	0.676782	0.676897	0.255945	0.495318
HCT, L/L	0.426660	1.000000	0.325570	0.405503
MCV, fL	0.427008	0.704503	0.677356	0.472510
MCH, fmol	1.000000	0.621995	0.704716	0.544441
MCHC, mmol/L	0.197243	0.849378	0.879559	0.969725
PLT, 10^12^/L	0.496130	0.733537	0.161184	0.544745

**Table 2 animals-12-03212-t002:** Results of Friedman’s ANOVA and post-hoc tests for the supplemented (G0) and control group (G1) in a blood hematological analysis.

Parameter, Unit	*p*-Value of G0	*p*-Value of G1
BASO%	0.01375	0.633762
EOS, 10^9^/L	0.017324	0.098309
EOS%	0.00569	0.060406
HTC, L/L	0.019976	0.056037
HGB, 10^12^/L	0.03228	0.052612
LYM, 10^9^/L	0.025522	0.032658
MCHC, mmol/L	0.007176	0.019034
MCV, fL	0.000099	0.000108
PLT, 10^12^/L	0.012166	0.948376
RBC, 10^12^/L	0.026264	0.024827

**Table 3 animals-12-03212-t003:** *p*-values representing differences between supplemented (G0) and control groups (G1) in a biochemical analysis.

Parameter, Unit	Day 1 G0 vs. G1	Day 28 G0 vs. G1	Day 56 G0 vs. G1	Day 84 G0 vs. G1
Albumin, g/L	0.677585	0.596702	0.520523	0.212295
AP, U/L	0.045155	0.014020	0.012612	0.037636
AST, U/L	0.241322	0.121225	0.017258	0.384674
Total protein, g/L	0.226477	0.820596	0.596702	0.405680
Bilirubin, µmol/L	0.212295	0.909722	1.000000	0.879829
Chlorides, mmol/L	0.733730	0.969850	0.241322	0.650148
Cholesterol, mmol/L	0.344705	0.241322	0.104111	0.104111
CK, U/L	0.384674	0.791337	0.850107	0.472676
P, mmol/L	0.762369	0.173618	0.909722	0.939743
GLDH, U/L	0.623177	0.850107	0.623177	0.762369
Glucose, mmol/L	0.096305	0.009109	0.025749	0.677585
GGTP, U/L	0.150928	0.762369	0.307490	0.241322
Creatinine, µmol/L	0.344705	0.307490	0.273037	0.650148
LDH, U/L	0.623177	0.075663	0.733730	0.449692
Mg, mmol/L	0.596702	0.879829	0.969850	0.075663
Urea, mmol/L	0.405680	0.969850	0.472676	0.449692
K, mmol/L	0.850107	0.596702	0.820596	0.791337
Na, mmol/L	0.969850	0.140466	0.449692	0.449692
TGL, mmol/L	0.705457	1.000000	0.069643	0.384674
Ca, mmol/L	0.570751	0.520523	0.405680	0.762369
Globulins, g/L	0.104111	0.545350	0.623177	0.520523
Albumin/globulin ratio, mmol/L	0.241322	0.405680	0.909722	1.000000
Lactic acid, mmol/L	0.256840	0.037636	0.879829	0.791337

**Table 4 animals-12-03212-t004:** Results of Friedman’s ANOVA and post-hoc test for the supplemented group (G0) and control group (G1) in a biochemical analysis.

Parameter, Unit	*p*-Value G0	*p*-Value G1
Albumin, g/L	0.006038	0.041825
AP, U/L	0.001032	0.001032
AST, U/L	0.011694	0.008724
Total protein, g/L	0.00477	0.000655
Bilirubin, µmol/L	0.036403	0.228919
Chlorides, mmol/L	0.000515	0.008724
Cholesterol, mmol/L	0.004844	0.059153
CK, U/L	0.012858	0.53987
P, mmol/L	0.032312	0.322733
GLDH, U/L	0.022853	0.000921
Glucose, mmol/L	0.001032	0.130501
GGTP, U/L	0.01919	0.000059
Creatinine, µmol/L	0.008724	0.000522
LDH, U/L	0.001531	0.000822
Mg, mmol/L	0.0000009	0.046223
Urea, mmol/L	0.001156	0.017809
K, mmol/L	0.017626	0.012858
Na, mmol/L	0.003467	0.000293
Ca, mmol/L	0.003467	0.01848
Globulins, g/L	0.01919	0.016287
Albumin/globulin ratio mmol/L	0.017626	0.000293
Lactic acid, mmol/L	0.143604	0.000237

## Data Availability

The data presented in this study are available on request from the corresponding author. The data are not publicly available for privacy reasons.

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
