# Peer review of "Evaluation of Shiitake Mushroom (Lentinula edodes) Supplementation on the Blood Parameters of Young Thoroughbred Racehorses"

_animals, 2022, doi:10.3390/ani12223212_

Round 1

Reviewer 1 Report

Please also refer to the appended marked up pdf of the submitted manuscript

1.       Avoid using general terms that poorly convey information

a.       Session: replace this term because it is incorrect and misleading. A single sampling time point is not a session. In both the simple summary and abstract it should be clearly stated that only one blood sample was taken at monthly intervals for 4 months.

b.       Line 25: it cannot be stated that “supplementation had an effect”. Only that there were differences in these measured variables ONLY at these few time points. This does not equate to an effect. A true effect would manifest, consistently, at least at time point 4. This did not occur.

c.       Line 30-31. “positive effect”? What was the positive effect? There was basically no effect, and certainly not a positive effect.

d.       Line 31: There is no evidence that supplementation “prevents sudden changes”. The lactate result in the control group is a serious outlier and the interpretation provided for this in the Discussion indicates lack of understanding of cellular metabolism.

e.       “morphology’ is incorrectly used. The term that should be used is hematology.

2.       Other Requirements

a.       Section 2.2 – provide content of bioactives deemed relevant for the observed differences.

b.       Section 2.4 – state how / why (statistically) 10 horse per group was used.

c.       Line 180: “differences were found” between what? Be specific. Different means greater? Less?

d.       What was the point of doing the Friedman’s Anova? And line 172. Differences between what? Over time? Between measures within group? Between groups? This is not clear in the methods, and the results are poorly presented. Are these results even needed?

e.       Replace Figures 1 and 2 with Tables showing means (not medians) and SDs AND that allows comparison between values for control and supplemented groups. The current presentation is physiologically and statistically meaningless.

f.        Line 246 – according to Figure 2 AP is greater in the supplemented group than control group – opposite of what is stated here. Also, the fact that this difference was present in sample 1 suggests a group difference, not a mushroom effect. You are over-interpreting the data.

g.       Line 274: you need to address why there is a median lactate value of nearly 8 mmol/L in horses at rest at ONLY this time point. This is NOT physiologically normal. The interpretation is seriously incorrect.

h.       Line 300 – training effect on albumin and blood volume – know the literature.

i.         Conclusions: the results and interpretations are overstated and misleading. These need to be corrected.

Reviewer 2 Report

The goal of this research was to determine the effects of feeding shiitake mushrooms to young horses. Overall this is interesting research – but the statistical analysis needs to be conducted differently before this paper can be considered for publication. Further, the authors need to change how the results are displayed.

The simple summary is generally well written, however a few changes need to be made:

-this reviewer suggests that the authors remove some of the scientific jargon such as “single statistical” at line 23.

-a brief description of the experiment that was performed would be good to set the reader up to understand what was tested and what the results were.

-the authors also need to make sure to be more specific in terms of what samples were measured in what tissue and when – this is hard to follow. I realize this is supposed to be a simple summary but the lack of detail makes it unclear.

The abstract needs to be edited to be more specific. The authors need to include P values and when discussing differences, they need to state which way the value changed – i.e. was it increased or decreased in the treatment group compared to the control?

-line 39: when were the blood samples collected?

-line 40: the sentence beginning here reads a little funny, re-word this

The introduction generally lacks flow – try to make it flow a little better and spend some time on the transition between the medical properties and the livestock species. The authors jump straight from medical properties to a weird one sentence paragraph on poultry and then right into one of their studies on horses. There needs to be more transitions here a little bit of text about why this is even important in horses. I would also suggest the authors spend less time describing their previous work and save that for the discussion. I also have a few specific comments listed below.

-line 62: in the sentence beginning at this line, the authors need to state what species these studies were completed in to give context to the reader. This is also true for the next sentence at line 64

-line 67: should read amino acids rather than amino acid

-line 69: change reduce to reduces

-Line 71: add the word the before skeletal system

-Line 72: It is strange to have a paragraph that is one system – try to work this into the rest of the introduction.

-Line 76: add a reference to this study

The materials and methods were overall well written and easy to follow. This reviewer did have a few concerns:

-were any measurements, such as weight or body condition score collected on the horses throughout the trial? Any measurements relating to their performance such as speed or VO2 max?

-The statistics section is hard to follow and needs to be re-written. Since the authors collected samples over time, the data needs to be analyzed as a repeated measure rather than at specific time points. In addition, random variable such as horse age should be included in the model.

A few other specific comments are listed below:

Line 110: do the authors mean 12.00 p.m?

Line 146: the information here was stated above at line 129. It is not necessary to state it twice.

The results are very hard to follow – the entire results section and the way that all of the data is displayed needs to be reconfigured. Specific comments regarding these changes can be seen below.

-In the paragraph beginning on line 179, the authors need to state what the P value is and make sure they are stating results clearly – this is hard to follow as written.

-In the paragraph beginning on line 184, it is not clear what the authors were testing – is this a repeated measures analysis? An average of the time points? More clarification is needed.

-In table 1 the P values do not need that many digits. It would also be nice to see directionality – which treatment group was increased or decreased? This reviewer would like to see actual values and then a P value as well rather than just P values.

-Table 2 needs to have actual values as well. This review is not sure what the authors are trying to show here – is this the P value for how these values changed over time?

-Figure 1 should be a paneled figure with a line graph and standard error bars. The way the data is presented here is hard to follow.

-Same comments as above for section 3.2 – the text in the results is hard to follow. The authors need to better describe the results, provide some directionality of the change and include that directionality and actual values in table 3.

-Same comments for table 4 as for table 2 above.

-Same comments for figure 2 as for figure 1 above

The authors need to better describe the results before the discussion can be thoroughly evaluated. Some specific comments have been added below.

-In the paragraph beginning on line 239 some directionality of the results would help with interpretation.

-The sentence beginning at line 260 has too much postulation – none of these things were measured. The authors need to make sure they are attempting to interpret the results, but not drawing conclusions that are beyond the scope of their data.

The conclusions section is somewhat hard to follow – the data looks like variable did change in the treated horses according to the P values? How are you comparing these?

Round 2

Reviewer 1 Report

The paper is improved, and the explanations helpful.

However, the authors are still overstating the interpretations and must refrain from making conclusions that go beyond what the data show.

The understanding of physiology appears to be generally weak, and for exercise physiology very weak. It reflects poorly on the authors when incorrect physiological interpretations are provided, and it also misinforms the reader. You need to get it right and cite papers that actually did the research to demonstrate mechanism, not weak descriptive papers.
